# Observations on aerosol optical properties and scavenging during cloud events

Antti Ruuskanen[1], Sami Romakkaniemi[1], Harri Kokkola[1], Antti Arola[1], Santtu Mikkonen[2,3], Harri Portin[1,*], Annele Virtanen[2], Kari E. J. Lehtinen[1,2], Mika Komppula[1], and Ari Leskinen[1,2]

[1]Finnish Meteorological Institute, P.O. Box 1627, 70211 Kuopio, Finland
[2]Department of Applied Physics, University of Eastern Finland, P.O. Box 1627, 70211 Kuopio, Finland
[3]Department of Environmental and Biological Sciences, University of Eastern Finland, P.O. Box 1627, 70211 Kuopio, Finland
[*]Helsinki Region Environmental Services Authority, P.O. Box 100, 00066 HSY, Finland

Correspondence to: antti.ruuskanen@fmi.fi

**Abstract**

Long-term statistics of atmospheric aerosol and especially cloud scavenging were studied at the Puijo measurement station in Kuopio, Finland, during October 2010 – November 2014. Aerosol size distributions, scattering coefficients at three different wavelengths (450 nm, 550 nm, and 700 nm), and absorption coefficient at wavelength 637 nm were measured with a special inlet system to sample interstitial and total aerosol in clouds. On average, accumulation mode particle concentration was found to be correlated with temperature with lowest average concentrations of 200 cm$^{-3}$ around 0°C increasing to 800 cm$^{-3}$ at 20°C. The scavenging efficiencies of both scattering and absorbing material were observed to have a slightly positive temperature correlation in in-cloud measurements. At 0°C temperature, the scavenging efficiencies of scattering and absorbing material were 0.85 and 0.55 with slopes of 0.005 $\frac{1}{°C}$ and 0.003 $\frac{1}{°C}$, respectively. Scavenging efficiencies were also studied as a function of the diameter at which half of the particles are activated into cloud droplets. This analysis indicated that there is a higher fraction of absorbing material, typically black carbon, in smaller sizes so that at least 20-30% of interstitial particles within clouds consist of absorbing material. In addition, the PM1-inlet revealed that approximately 20% of absorbing material was observed to reside in particles with ambient diameter larger than ~1 μm at relative humidity below 90%. Similarly, 40% of scattering material was seen to be in particles larger than 1 μm. Altogether, this dataset provides information on the size dependent aerosol composition and in-cloud scavenging of different types of aerosol. The dataset can be useful in in evaluating how well the size dependent aerosol composition is simulated in global aerosol models and how well these models capture the in-cloud scavenging of different types of aerosol in stratus clouds.

**1 Introduction**

Aerosol particles in the atmosphere induce radiative forcing by absorbing and scattering light (direct effect) and by modifying cloud properties by acting as seeds for cloud droplets (indirect effect) (Lohmann and Feichter, 2005). The interaction of aerosols with radiation depends highly on their physico-chemical properties, such as size, morphology, and chemical composition. These properties are source-specific and evolve during the life-cycle of aerosol in the atmosphere. Aerosol particles can grow by condensation, coagulate with each other, undergo phase change, or get cloud-processed, i.e., grow in clouds due to chemical processes and coagulation until they are finally removed from atmosphere through dry or wet deposition. If the atmospheric lifetime of an aerosol particle is long, it can travel long distances from the emission source regions and cause radiative forcing over areas with low natural aerosol sources, like for example black carbon (BC) over Arctic areas (Chaubey et al., 2010, Latha et al., 2005). To estimate the radiative forcing of the anthropogenic aerosol, all these processes from aerosol formation to their removal must be understood and resolved.

Although aerosol properties may change already at relative humidity (RH) below 100% due to microphysical processing, aerosol properties undergo a much faster change within clouds where a fraction of submicron aerosol particles is grown into cloud droplets with sizes of the order of 10 micrometres in diameter (Väisänen et al. 2016, Anttila et al. 2012). In order to be cloud-processed, an aerosol particle must be able to act as a cloud condensation nucleus (CCN). The CCN activity of an aerosol particle depends on its size and hygroscopicity (Swietlicki et al., 2008). Inorganic salts, such as sulphates, nitrates, and ammonium, are known to be hygroscopic while e.g. pure black carbon is highly non-hygroscopic (McMeeking et al., 2011). Typically, hygroscopic particles such as sulphate are known to be efficient scatterers of light, whereas insoluble compounds such as BC absorb light efficiently (e.g., Bergstrom et al. 2002). Therefore, in the case of particles of the same size, the scattering aerosols are scavenged by cloud droplets more frequently, and this has also been observed in several studies (Browse et al., 2012; Berkowitz et al., 2011; ; Verheggen et al., 2007; Cozic et al., 2007; Komppula et al., 2005). On average, if cloud formation leads to wet scavenging of aerosol, this results in less numerous, more absorbing, and smaller aerosol particles. As a result, the direct cooling effect through light scattering is decreased compared to the warming effect of absorbing aerosols.

The optical and CCN properties of BC containing particles are influenced by their size and the mixing state. Fors et al. (2011) and Väisänen et al. (2016) found that in externally mixed ambient aerosol the accumulation mode typically contains a higher fraction of hygroscopic particles than the Aitken mode. Due to aging processes, atmospheric BC particles contain a variying fraction of other chemical components (inorganics, water, water-soluble organics), and these compounds may enhance the light absorption of the particles (Bond et al., 2006). Coating of BC may also increase the CCN activity through elevated hygroscopicity (Liu et al., 2013) although the enhancement depends on the amount of coating and is not always extensive (Cappa et al., 2012). However, because of such a mixing, it is not straightforward to determine if the overall role of BC containing particles in the atmosphere is always warming, or if these particles cause negative forcing through the Twomey effect.

Experimental information about physical, optical, and chemical properties of aerosols is achieved from long-term in situ measurements which have been and are being conducted in several places throughout the world (e.g., Collaud Coen et al., 2013; Asmi et al., 2013). The effect of clouds or precipitation on aerosol optical properties has also been studied in several campaigns (Zhang et al., 2012; Berkowitz et al., 2011; Hyvärinen et al., 2011; Chaubey et al., 2010; Marcq et al., 2010; Yamagata et al., 2009; Cozic et al, 2007; Latha et al., 2005) and by conducting model calculations (e.g., Browse et al., 2012; Croft et al., 2009). Wet scavenging of different compounds has significant importance on global aerosol composition. As scavenging of different compounds depends on where the compounds reside in the aerosol size spectrum, there is a need for

measurements on how the chemical compoounds are partitioned between the interstitial aerosol and cloud droplets. However, such long-term measurements are conducted only in few locations related to low altitude clouds or mountain stations. For example, a handful of studies have been carried out in which cloud scavenging efficiency of black carbon has been studied as reviewed by Yang et al. (2019).

The partitioning of aerosol particles between the interstitial and activated (i.e. cloud droplet) phase can be studied by parallel measurements of the interstitial and cloud activated aerosols. In such studies, different methods have been used: an interstitial inlet with a cut-off diameter low enough to remove cloud droplets, a counterflow virtual impactor (CVI) to extract cloud droplets (Noone et al., 1988) and ice particles from mixed phase clouds (Mertes et al., 2007), and a total air inlet, which samples both the interstitial and activated phases (Weingartner et al., 1999). In principle, the most suitable instrument to study the properties of activated aerosols is the CVI, which samples only large particles (or cloud droplets) with enough kinetic energy to pass through the counterflow section that removes small (interstitial) particles, and enables to study the activated fraction directly. A drawback of CVIs is the enrichment of the sample, as the input flow is larger than the sample flow (Twohy et al. 2003). The total and interstitial inlets sample particles smaller than their predetermined cut-off sizes which enables us to get information about both cloud scavenged and non-scavenged particles. It must be noted that since the two-inlet method relies on subtracting the properties of interstitial aerosol particles from those of total aerosol particles, some discrepancies may be seen if the studied property is not of additive in its nature, or if there is a delay between the interstitial and total aerosol measurements, particularly in fast changing situations. Furthermore, aerosol particles may start growing already below 100% RH due to water uptake and other microphysical processing (Ervens et al., 2011 ;Wonaschuetz et al., 2012), which may lead to a situation where a fraction of submicron aerosol particles is grown into cloud droplets with sizes of the order of several micrometres in diameter (Väisänen et al. 2016; Anttila et al. 2012) and are thus removed by the interstitial inlet and regarded as cloud scavenged although this is not the case. All in all, each of these systems has its pros and cons, and in finding a suitable method one must bear in mind the needs and the restrictions of the site and the sampling setup.

In this work, we used a combination of interstitial and total inlets, and analysed the properties of activated particles as a difference between the two inlets. Beyond reporting typical aerosol properties, our aim was to clarify how ambient conditions and properties of aerosol population influence the scavenged fraction of light absorbing and scattering aerosol particles.

## 2 Methods

### 2.1 Site description

The measurement site is located on the top of an observation and retransmitting tower at Puijo (62°54'34" N, 27°39'19" E) in Kuopio, Finland. The measurement height is 306 m above sea level and 224 m above the level of the surrounding lake. In the surroundings of the measurement site there are a few local pollutant sources, such as a paper mill, a district heating plant, traffic routes, and residential areas with biomass-fired combustion appliances. At this measurement site we can study how the properties of the fresh aerosol emissions from the local sources differ from aged aerosol observed at Puijo. Furthermore, the measurement site is inside low-level clouds approximately 8% of the time, which enables us to study the aerosol-cloud interactions in a continuous, long-term manner. The cloudiest period is during autumn (Fig. 1), when the tower is inside clouds about 13% of the time.

Approximately 91 000 people live in the urban area of Kuopio (Fig. S1). The Puijo hill (elevation 150 m) and the measurement station are located ca 2 km NW from the city centre. There are a few well-known local sources: the district heating plant 3.5 km SE, the paper mill 5 km NE, and a highway across the city. The distances and directions are in relation to the measurement station. Several residential areas are located nearby, but mostly east and south of the station. A more detailed description of the measurement site, the surroundings, and previous studies are given in Leskinen et al. (2009, 2012), Portin et al. (2009, 2014), Hao et al. (2014), Väisänen et al. (2016) and Romakkaniemi et al. (2017). An analysis of aerosol source areas is conducted in Väisänen et al. (2020)

## 2.2 Instrumentation

We measured temperature, relative humidity, horizontal wind direction, and visibility on the roof of the tower with a time resolution of 1 min, by using a temperature and humidity transmitter (Vaisala, Model HMT330MIK), an ultrasonic wind anemometer (Thies, Model UA2D), and a present weather sensor (Vaisala, Model FD12P). We also measured the cloud base height at the nearby Savilahti weather station (2 km south-west from Puijo) with a ceilometer (Vaisala CT25K). An overall distribution of selected meteorological data over the calendar months of the measurement period is given in Fig. S2.

Aerosol measurement instruments were placed on the top floor inside the tower. The sample air for the instruments was drawn through two separate sampling lines: the interstitial and the total aerosol lines. The former is equipped with an impactor (Digitel DPM 10/01/00/16) whose cut-off size is 1.0 µm at its nominal flow rate of 1.0 $m^3$ $h^{-1}$ (16.7 l $min^{-1}$). The impactor removes coarse particles and cloud droplets from the sample, and it has been characterized and calibrated by the manufacturer. The latter sampling line, in turn, has a snow hood and a heated inlet, with a cut-off size of 40 µm, characterized by Weingartner et al. (1999). Water on the cloud droplets evaporates in the total aerosol inlet and sampling line. The resulting total aerosol contains both the interstitial aerosol and cloud droplet residual. The properties of the cloud droplet residual can then be estimated, at least for additive parameters, by subtracting the property of the interstitial aerosol from that of the total aerosol. This so-called twin-inlet system makes it possible to compare the properties of non-activated and activated aerosols, making it a suitable method for studying how aerosol mixing state affects cloud droplet formation and aerosol scavenging. The inlet system has been characterized by comparing the interstitial and total particle size distributions during a cloud-free weather and they have been found, after particle loss corrections, to match well in the measured range of 3–800 nm of the differential mobility particle sizer (DMPS) described below.

We measured the aerosol scattering (at 450, 550, and 700 nm) coefficients by using a TSI Model 3563 integrating nephelometer (Anderson et al., 1996) and absorption (at 637 nm) with a Thermo Model 5012 multi-angle absorption photometer (MAAP; Petzold and Schönlinner, 2004). Both instruments took samples from both the interstitial and total line with the help of a controllable valve system (shown in Fig. 2). The system switched the sampling lines in 6-min intervals. In this configuration, the flow rate through both instruments was 8.0 lpm, and we collected the data with a time resolution of 1 min from both devices.

Particle number size distributions were measured using a twin DMPS (self-constructed) (Winklmayr et al., 1991; Jokinen and Mäkelä, 1997). The system was also used to get information about cloud activation of the particles and to determine the diameter ($D_{50}$) at which half of the particles of that size are activated into cloud droplets. The measured size range was originally 7–800 nm and was extended to cover the range of 3–800 nm in February 2012. DMPSs were also part of the controllable valve system together with the nephelometer and MAAP. Throughout the measurements, we performed periodic flow checks and calibrations for the instruments to ensure the high quality of the measurements.

The overall raw data coverages for the nephelometer, MAAP, and DMPS were 64%, 95%, and 95%, respectively. After removing invalid data caused by instrument malfunctions, contaminations, and periodic calibrations and flow checks, the valid data coverages were 35–85%. More detailed, monthly valid data coverage for each instrument is illustrated in Fig. 1b.

## 2.3 Data processing

The analysed data set covered the period 5 October 2010 – 30 November 2014, including meteorological variables, and aerosol particles optical properties and size distributions for both the total and interstitial lines. All data was averaged for 12 minutes based on DMPS time resolution. First, we ruled out non-usable data during maintenance, calibrations, flow checks and autozeroing (i.e., periodic automated measurement to determine the background noise) from the optical variables. In addition, data with abnormal peaks and negative values was removed. As in our earlier work (Leskinen et al., 2012), we
omitted the scattering data when relative humidity (RH) exceeds 50% in the integrating nephelometer inlet and corrected the scattering coefficient for truncation errors (Anderson and Ogren, 1998).

We categorized the data into three categories according to the cloudiness conditions and named them as "no cloud", "cloud", and "intermediate". The criteria for "no cloud" were: RH < 80% and visibility > 8000 m or, if either RH or visibility data
was not available, cloud base height (CBH) > 500 m. For "cloud" category: the visibility limit was < 200 m or, if visibility data was not available, measured RH > 99%. The periods not meeting either of these criteria were labelled as "intermediate". The fractions of each category from 5 October 2010 to 30 November 2014 were 44.6%, 8.0%, and 47.4% for "no cloud", "cloud", and "intermediate", respectively.

We calculated scavenging fractions, i.e., the fractions of the absorbing and scattering material in the activated (cloud) phase with the following equations:

$$F_{scav,a} = \left( \frac{\sigma_{ap,\lambda,tot} - \sigma_{ap,\lambda,int}}{\sigma_{ap,\lambda,tot}} \right) \tag{1}$$

and

$$F_{scav,s} = \left( \frac{\sigma_{sp,\lambda,tot} - \sigma_{sp,\lambda,int}}{\sigma_{sp,\lambda,tot}} \right). \tag{2}$$

In Equations (1) and (2) $\sigma_{sp,\lambda}$ is the scattering coefficient and $\sigma_{ap,\lambda}$ is the absorption coefficient at a given wavelength $\lambda$.
Subscripts *int* and *tot* refer to the interstitial and total lines.

In this study we defined the accumulation mode number concentration ($N_{acc}$) as the number concentration of particles larger than 100 nm in diameter (electrical mobility diameter). $N_{acc}$ is found to be a representative proxy for CCN in boundary layer stratified clouds in boreal conditions (Lihavainen et al. 2008). The total number concentration ($N_{tot}$) is defined as the number concentration of all particles ranged from 3 or 7 nm to 800 nm (see section 2.2). In addition, we calculated volume
concentrations ($V_{tot}$ and $V_{acc}$) with the same particle diameter limits.

## 2.4 Observed clouds

Due to the low altitude of the observation location, all clouds are stratiform and capped with a temperature inversion. The criterion for an average visibility of less than 200m over the 12-minute period effectively excludes all broken clouds. The
average cloud base height, estimated from the data of the nearby ceilometer, is 170 m, which would turn into an average (adiabatic) liquid water content of ~0.06 g m$^{-3}$ that is comparable to the average of ~0.05 g m$^{-3}$ obtained from cloud droplet size distribution observations at Puijo (Portin et al., 2009). At Puijo, we do not have continuous and reliable information on the cloud physical depth or possible clouds residing above the observed, lowest boundary layer cloud, which could affect the

radiative cooling of the lowest cloud top and subsequently the boundary layer dynamics (Leung et al., 2016). However, we used the present weather sensor data for separating precipitating and non-precipitating, and we observed that removing the precipitating clouds had a negligible effect on the results because we studied instantaneous cloud scavenging instead of total wet scavenging. Thus, the data is also applicable to cases with precipitation, which amounts 24 % of the cloud data.

## 3 Results and discussion

### 3.1 Aerosol number concentrations

The aerosol emissions near the Puijo site have a large annual variability as the contribution of different emission sectors to the aerosol load is highly dependent on the season. To analyse the characteristics of size dependent aerosol properties in different weather conditions, we studied the behaviour of total and accumulation mode particle number concentrations as a function of temperature in all measured cases (Fig. 3a,b,c) and in-cloud cases (Fig. 3d,e,f). Panels a and d present total number concentrations, panels b and e present accumulation number concentrations, and finally, panels c and f present ratio of the accumulation and total number concentrations. Note the different concentration ranges in each panel and lack of cloud event data in temperature ranges from -20°C to -15°C and from 20°C to 25°C.

The total number concentration has a correlation to air temperature as seen from the all data case (Fig. 3a) and from the cloud only cases (Fig. 3d). However, at low temperatures the total particle number concentration increases (Fig. 3a) which can be caused by increased local anthropogenic emissions. The main reason during cold weather is the need of heating and electricity whereas the effect of biogenic emissions is no longer relevant (Geron and Arnts 2010, Tarvainen et al. 2005, Leaitch et al. 2011, Paasonen et al. 2013). In addition, it has been shown that vehicle emissions have a higher impact in rather cold than warm temperature (Wang et al. 2017). While most of the time total number concentrations are relatively low, aerosol nucleation events with high total concentration occur, and this can be seen as the difference in median and average concentrations.

We can see a parabolic curve with a minimum around 0°C (Fig. 3b) in the number concentration of accumulation mode particles presented as a function of temperature (all data case). Similar characteristics in accumulation mode number concentrations have also been reported in previous studies (Asmi et al. 2016, Paasonen et al 2013, Hussein et al. 2006). In warm temperatures (above +10°C), the accumulation mode particle number concentration increases rapidly which is likely caused by an increase in biogenic emissions (Ahlm et al. 2013). Additionally, wet deposition tends to decrease in extremely warm temperatures as precipitation decreases. However, the low concentrations at approximately 0°C temperatures are generally thought to be caused by lack of biogenic emissions (Heikkinen et al. 2020) as well as an increased amount of precipitation, and thus, wet deposition. Interestingly, the number concentration begins to increase towards colder temperatures in the all-data case but not in cloud event case. The increase could be explained by decreased amount of deposition and lower height of the boundary layer together with an increasing amount of aerosol from local house heating and energy production.

In addition to total and accumulation particle number concentrations, we studied the fraction of accumulation sized particles out of the total number. The ratio of accumulation to total particle number concentrations of all data (Fig. 3c) and cloud case data (Fig. 3f) show a similar parabolic curve as in Fig. 3b. However, the curve is less pronounced in cloud case data, especially at cold temperatures. This backs up the notion made in the previous paragraph about accumulation particles. Precipitation can be one of the explaining factors since precipitation events are most frequent around 0°C. Another interesting notion is that the ratio is slightly higher on average during cloud cases, excluding the below -5°C region. This

would imply faster coagulation/coalescence removal of nucleation and Aitken mode particles than that of accumulation mode particles (Laakso et al., 2003).

## 3.2 Influence of environmental parameters on the aerosol optical properties during cloud events

### 3.2.1 Relative humidity and scavenged particle volume fraction

5   Information about the composition size distribution, especially with respect to scattering and absorbing compounds, can be inferred by how their sampling in the interstitial line changes at different humidities. The compounds which are dominant in the largest particles are scavenged at high relative humidities even without any clouds present. These large particles are not considered to be cloud scavenged. Meanwhile, the scavenging of the compounds in smaller particle sizes is affected less by the increase in the relative humidity.

Here we present the scavenging efficiencies of scattering and absorbing material studied as a function of air relative humidity (Fig. 4 a,c) and scavenged total aerosol volume (Fig. 4 b,d). The box plots in Fig. 4 a and c illustrate how the scavenged fraction changes when there is a transition from cloud free to cloudy periods. The relative humidity subplots show an interesting effect of scavenging even at moderate air humidity. Approximately 40% of scattering and 20% of absorbing

15   material is scavenged by the impactor already at 50% RH. However, there is a large variability in the observed scavenging fractions because of different aerosol sizes and chemical compositions during the whole measurement period. This can be seen from the percentiles in the box plots. Interestingly, on average 20% of aerosol absorption is caused by particles close to or bigger than 1 µm. The origin or composition of this aerosol is not known as it is larger than typically analysed with mass spectrometry for example. However, the fraction is very similar to the fraction of absorbing material in particles between 1

20   µm and 10 µm in Hyytiälä, Finland as reported previously by Luoma et al. (2019). When reaching 100% air humidity, the scavenging efficiencies of both scattering and absorbing material increase. This is the transition zone to cloud cases. On average 70% of scattering material but only 30% of absorbing material is scavenged when the cloudy period begins.

While Figure 4 a and c showed the change in the fraction of scavenged scattering and absorbing material in transition from

cloud free to cloudy conditions, Figures 4 b and d illustrate how the fraction changes further when the fraction of activated particles increases. Even in the wavering cloud cases, 70% of scattering and nearly 30% of absorbing material is scavenged by clouds. As expected, in more prominent cloud cases less scattering and absorption is measured from the interstitial sample line, which leads to higher scavenging fractions. In practice, all of the scattering and nearly 80% of absorbing material is scavenged by clouds when more than 90% of submicron aerosol volume is scavenged.

These results indicate a couple of notions regarding the sampling setup and data analysis. First, a sampling system with 1.0-µm diameter cutoff can lose relevant particles (measurement site specific). Depending on the ambient conditions, hygroscopic and aged BC particles, that are expected to be sampled, are undetected by the interstitial line due to particle growth. Second, scattering is dominated by large particles, which is expected by scattering theory and is also reported in

earlier studies (Anttila et al. 2012, Portin et al. 2014). However, the scattering data can greatly help in explaining the absorption data. As mentioned earlier, approximately a 40% difference between the sampling lines even at moderate air humidity implies the presence of larger than 1 µm particles or growth of hygroscopic particles beyond 1 µm in size. The same applies to the absorbing material that has aged and is coated with hygroscopic coating (e.g., aged BC particles) and can be removed before cloud formation. However, BC typically resides in the particles that are much smaller than 1 µm and thus

the gradient in scavenging efficiency as a function of relative humidity is less steep than that for the scattering fraction. Another indication for absorbing material to reside in the smaller sizes is that more than 20% of the absorbing material is not scavenged after more than 90% of aerosol volume is scavenged by clouds.

It is clear from Fig. 4 that the dispersion of the data is fairly large and includes non-physical values such as scavenging efficiencies out of the range from 0 to 1. This cannot be avoided as there is a slight time difference between the measurement from the interstitial and total line, and this can introduce negative scavenging efficiency values when considering small absolute values of scattering and absorption coefficients. Another reason for the anomalous values is that the interstitial line values are practically zero in many of the cloud cases but, due to the measurement accuracies, can sometimes produce small negative values (especially for the scattering coefficient). This leads to situations where the estimated scavenging fraction is slightly over unity. The third and last possible source of dispersion resides in long term measurements. Since the dataset is large, it is probable that similar kinds of meteorological conditions occur multiple times during the measurement period but with different kinds of aerosol properties.

In comparison to Cozic et al. (2007) and Motos et al. (2019) we did not observe 1:1 line for scavenged absorption matter as a function of scavenged particle volume. It is worth noting that in our case the scavenged aerosol volume is calculated from the DMPS size distribution, thus including only submicron particles. However, while this has been taken into account, our results show that some amount of absorbing matter remains in the particle phase even with high particle volume scavenging efficiency. This difference is presumably caused by the vicinity of the BC source regions, and relatively low supersaturation in the observed clouds which limits the activation of small particles.

### 3.2.2 Impact of air temperature

One of the main objectives of this study was to investigate how the optical properties of aerosols change during cloud scavenging events at different temperatures. In Fig. 5, cloud scavenged fractions of scattering (a) and absorbing (b) material are presented as a function of temperature with boxplots. The boxes have a fixed width of 5°C ranging from -15°C to 20°C.

We can see from Fig. 5 that there is a correlation between temperature and scavenging efficiencies of both scattering and absorbing aerosols. Linear fits were calculated to estimate the connection. As Pitkänen et al. (2016) and Mikkonen et al (2019) showed, the errors in measured variables need to be taken properly into account in the linear fits. Thus, the slopes were calculated with Deming-regression (Deming, 1943). The slopes for scavenging efficiencies were calculated to be approximately 0.0052 (95% CL for slope: 0.0042-0.0062) for scattering (Fig. 5a) and 0.0034 (0.0022-0.0046) for absorption (Fig. 5b). Which means that the effects are statistically significant and corresponding to approximately 5 percentage unit change for scattering and 3 for the absorbing material in the scavenged fraction at 10°C change in temperature, respectively. The efficiencies were observed to be 0.85 and 0.55 at 0°C for scattering and absorbing matter.

In both Fig. 5a and 5b a clear decrease in scavenging efficiency from 0°C to -15°C can be seen. In the case of absorption, one explaining factor can be the increasing amount of fresh BC emissions from the local heating sources. Freshly emitted BC particles are known to be hydrophobic. However, since the effect is also seen in the scattering data, it is likely that there are some processes that prevent cloud activation. For example, when there are mixed phase clouds, less numerous ice particles could cause the more numerous liquid cloud droplets to evaporate due to lower saturation vapour pressure over ice. On the other hand, mixed phase clouds are not common in these temperatures. It is also possible that in colder conditions the radiative cooling from the cloud top is weaker and thus the negative buoyancy driving boundary layer mixing is also weaker. This would lead to lower updrafts at the cloud base thus limiting the activation of cloud droplets. To study this further would require modelling exercise utilizing for example Large Eddy simulations model with detailed inputs for atmospheric vertical profiles and aerosol conditions. However, this is beyond the scope of current study both due to lack of profiling data and computational resources it would require for observational period of several years.

When considering temperatures above 0°C we can see a rather clear increase in the scavenging efficiency of scattering material. This could be explained by the increase of available water, increasing amount of hygroscopic material in particles, e.g., from biogenic sources, and a general increase in particle size which can also likely be attributed to condensation of semi- and non-volatile biogenic compounds on pre-existing particles. Over the boreal region, biogenic compounds do have a significant effect on the aerosol load during the warm months and their emissions increase with increasing temperature (Liao, et al., 2014). Overall, an increase in the biogenic contribution, decrease in the anthropogenic contribution (BC), and the aging of long-range BC increases the scavenging efficiency of scattering material.

Our results are qualitatively similar to the findings of Cozic et al. (2007) about the effects of temperature on the scavenging efficiency of absorbing material. At 0°C, the scavenging efficiency of black carbon was found to be 0.6 and it was seen to decrease towards colder temperatures. However, in our case the decrease in the scavenging efficiency is not as strong as the decrease reported by Cozic et al. (2007). In another study (Schneider et al. 2017) an average scavenging efficiency of 0.24 was observed over 14 campaigns with mean ambient temperatures varying from -3.0°C to 9.2°C. On the other hand, in each study the impactors for differentiating interstitial particles were different. In our study, we used a PM1 impactor while Cozic et al. (2007) and Schneider et al. (2017) used PM2.5 and PM5 impactors, respectively. These could slightly affect the results as a higher cut-off could, on average, lead to smaller scavenging efficiency when the smallest cloud particles might be counted as aerosol particles.

### 3.2.3 Impact of particle activation size

To obtain more information about the cloud activation potential of different sized particles in different weather conditions, we studied how the $D_{50}$ values behave in different conditions. Fig. 6 illustrates $D_{50}$ as a function of temperature (Panel a) and as a function of the scavenged fraction of the total volume (Panel b). Additionally, a fraction of scavenged scattering (Panel c) and absorption (Panel d) are presented as a function of $D_{50}$. 5$D_{50}$ has a local minimum at approximately 0°C (Fig. 6a) which matches the peak in the scavenging efficiency of the absorbing matter (Fig. 5b). This can be expected as also $N_{acc}$ has a minimum value close to 0°C, and thus lower condensation sink during the activation process allows higher supersaturation.

Considering that Fig. 6b we can see that the $D_{50}$ value seems to saturate at low fractions of scavenged aerosol volume. This phenomenon can also be seen in Fig. 4b and 4d. Excluding the saturated part, the $D_{50}$ value decreases rather linearly with the scavenged aerosol volume. This is logical; as more of the aerosol population volume is scavenged, the smaller and smaller particles get scavenged into cloud droplets. It must be noted that the 1 μm cut-off impactor in the interstitial line may also influence the analysis. For example, a hygroscopic interstitial aerosol particle larger than 250 nm may grow larger than 1 μm due to the uptake of water within a cloud and may thus be removed by the impactor and counted as an activated particle. This may be true, e.g., in the case of a radiation fog, where activation diameters of up to 364–450 nm have been observed (Hammer et al. 2014). In addition, at our station, some of the clouds observed can be well-developed fogs. In our case, however, there are only a few observations in the unclear range where $D_{50}$ is affected by the interstitial inlet and thus the long-term statistics are not affected.

Scavenging efficiencies of scattering and absorbing material as a function of $D_{50}$ support the idea that absorbing matter tends to be in relatively small sized particles and most of the scattering is caused by large particles as mentioned in 3.2.1. This can be seen as high scavenging efficiency of scattering material even at $D_{50}$ of >300 nm while scavenging efficiency of absorbing material is low. In addition, only 70% of the absorbing material is scavenged with $D_{50}$ values higher than 50 nm.

Although, qualitatively the finding is similar to Motos et al. (2019) in the sense that more material is scavenged when $D_{50}$ is small, which in turn corresponds to high supersaturations. However, our results differ to those of Motos et al., (2019) in the sense that in our study, some of the BC is too small to be scavenged. This can be due to both fresher and less CCN active aerosol emissions close to Puijo, or, lower supersaturations typical of low altitude clouds observed at the Puijo station. We further studied if there is a difference in the partitioning of BC on different sized particles in different temperatures by allocating Figure 6d data on different temperature bins like Figure 5b (Supplementary Figure 3). However, no clear temperature correlation was observed, thus suggesting the relevant role of nearby emission in cloudy conditions to be independent of air mass origins.

**4 Conclusions**

We investigated the size dependence of scattering and absorbing material in different conditions with a measurement setup which samples hydrated aerosol from two inlets. One inlet samples all-sized aerosol particles and droplets and the other particles that can be assumed to remain inactivated within a cloud (i.e., interstitial particles). The main findings can be summarized as:

1) The measurements showed that aerosol number concentrations correlated with temperature . The total particle number concentrations increased with temperature likely due to an increase in the amount of biogenic aerosol. However, accumulation mode number concentrations exhibited a parabolic type correlation to temperature, reaching minimum at approximately 0°C. The increase in the concentration with temperature is due to an increase in the amount of biogenic aerosol, and the increase in sub-zero temperatures is caused by an increase in aerosol emissions from local sources due to residential heating and inversion conditions limiting the vertical mixing.

2) With measurement setup, we observed that a fraction of scavenged scattering and absorbing material remains constant up to 80% to 90% relative air humidity, above which their scavenged fraction increases. For scattering aerosol, this increase in the scavenged fraction initiates at a lower RH compared to absorbing material. This indicates that a higher fraction of absorbing material resides in smaller particles. However, on average 20% of absorbing material is found in particles larger than 1 μm in diameter.

3) In cloud cases, the measurements of the scavenged fraction of scattering and absorbing material showed the same tendency of absorbing material to reside in the smaller particles. The scavenged fraction of scattering material was found to be high for the whole range of $D_{50}$ values, while the scavenged fraction of absorbing material reached 70% only for the smallest $D_{50}$ values of approximately 50 nm. A subtle linear temperature correlation was observed for average scavenging efficiencies of both scattering and absorbing aerosol. At 0°C temperature, the scavenging efficiencies were 0.85 and 0.55 for scattering and absorbing matter, respectively. The slopes were approximately $0.005 \ °C^{-1}$ and $0.003 \ °C^{-1}$ .

The observations of $D_{50}$ offer a useful dataset for large scale aerosol modellers to evaluate the aerosol-cloud interactions in the models as they provide size dependent information about aerosol activation to cloud droplets. Data for such validation is scarce and, for example, aerosol-cloud interaction parameters from satellite observations are highly derived and include large

uncertainties (e.g., Grosvenor et al., 2018). Especially, wet deposition of absorbing material in global scale models can be evaluated using this dataset. The methods used in this study could be also applied in different kinds of in-situ sites. Overall, the data presented is especially useful for evaluating the cloud scavenging parameterisations in atmospheric models as it provides information on size dependent cloud scavenging efficiency of both absorbing and scattering material. It also

provides long term statistics on the concentration of CCN relevant particles at different temperatures.

**Data availability**

Data is available upon request from the authors.

**Author contribution**

AR, AL, SR, and MK planned the study, AR, AL, and SM carried out the data analysis, and AR prepared the manuscript with contributions from all co-authors.

**Competing interests**

The authors declare that they have no conflict of interest.

**Acknowledgements**

This work has been supported by the Maj and Tor Nessling Foundation for PhD thesis work. In addition, SR and HK acknowledge the Finnish academy (project nos. 283031, 285068 and 309127, the Centre of Excellence in Atmospheric Science, no. 307331), and Horizon 2020 Research and InnovationProgramme under Grant Agreement 821205 (FORCeS) for funding.

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

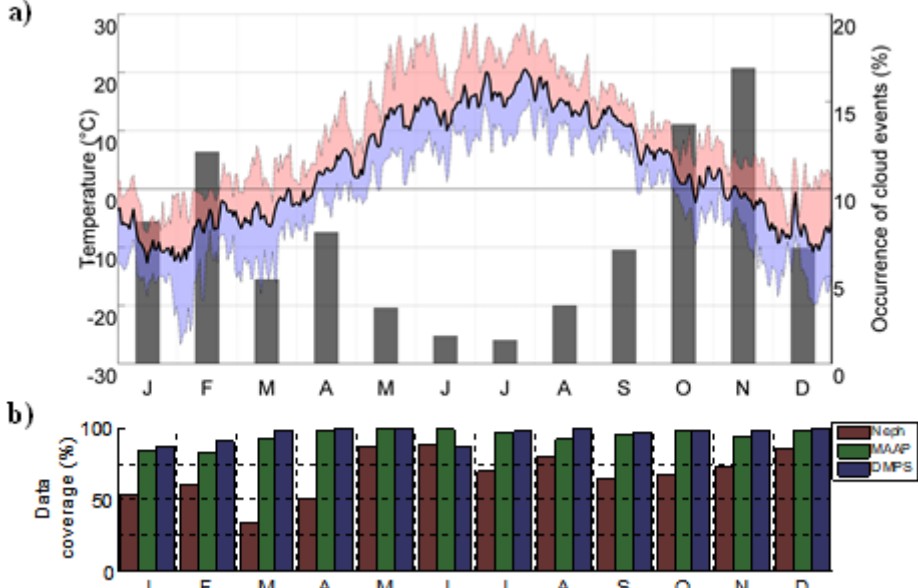

Figure 1: a) Average daily temperatures with minimum and maximum values over the studied period (left vertical axis) with occurred cloud events frequency (right vertical axis). The average temperature is presented with the black line and the daily minimum and maximum temperatures are presented with the lines associated with blue and red areas, respectively. The
5   occurrence of cloud events is presented with grey bars. b) Data coverage of nephelometer (red), MAAP (green), and DMPS (blue) during the cloud event periods. The coverage includes also values below device detection limits.

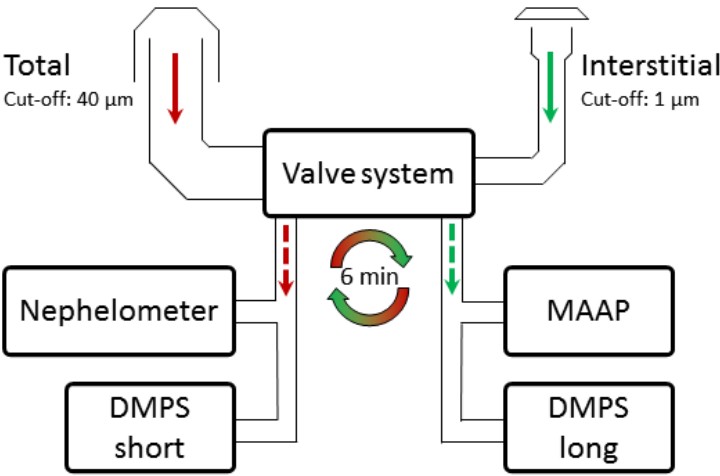

**Figure 2: Schematic figure of the sampling line and valve system. Total (red) and interstitial (green) samples reach valve system, where the samples are directed to the devices. Valve system switches the lines between two stages: in stage one a pair of devices sample from total line and in stage two the same pair samples from interstitial line. Naturally the other pair of devices samples vice versa.**

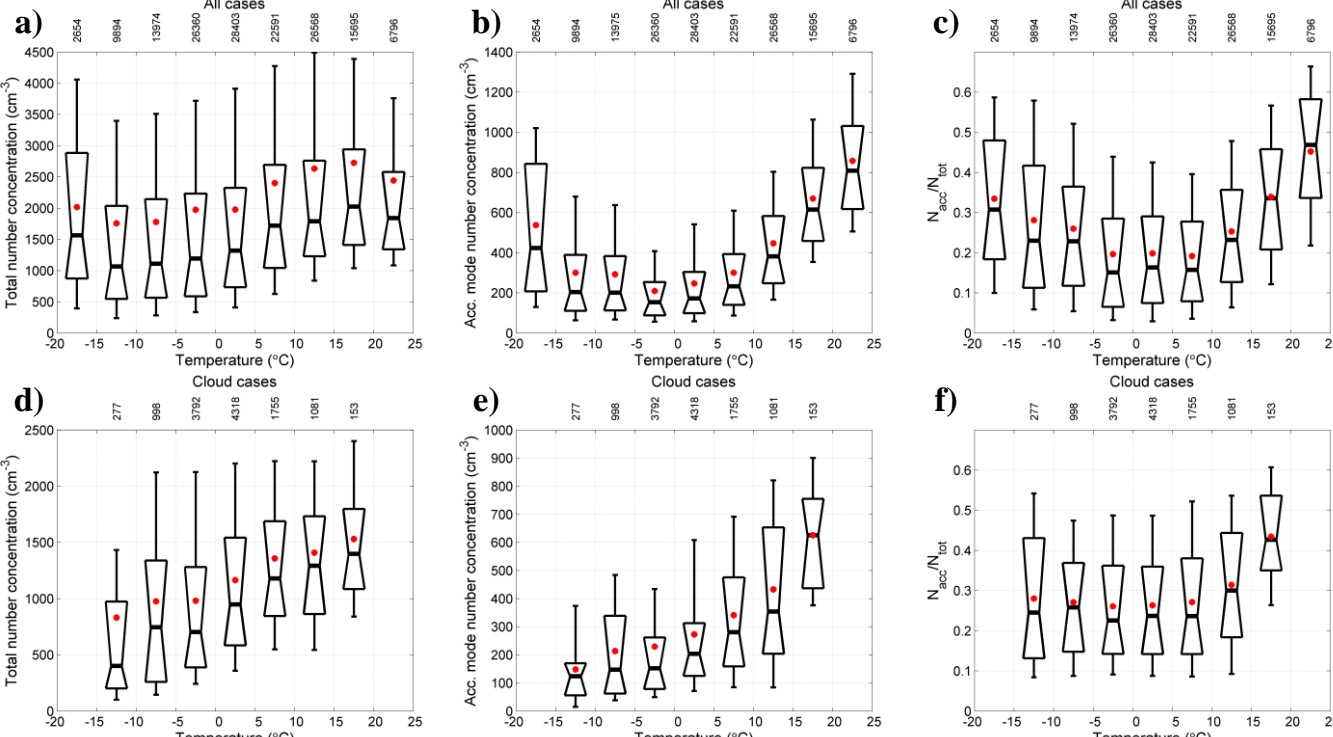

**Figure 3: Box plots of (a) total number concentration, (b) accumulation mode number concentration, and (c) ratio of accumulation and total number concentrations in all measured cases. Similarly presented plots of (d) total number concentration, (e) accumulation mode number concentration, and (f) ratio of accumulation and total number concentrations in cloud cases. The boxes have 10th, 25th, 50th, 75th, and 90th percentile marked with black lines. Mean of each box is marked with red circle. Data points per box are presented above the boxes. All subfigures (a)-(f) are presented as a function of temperature.**

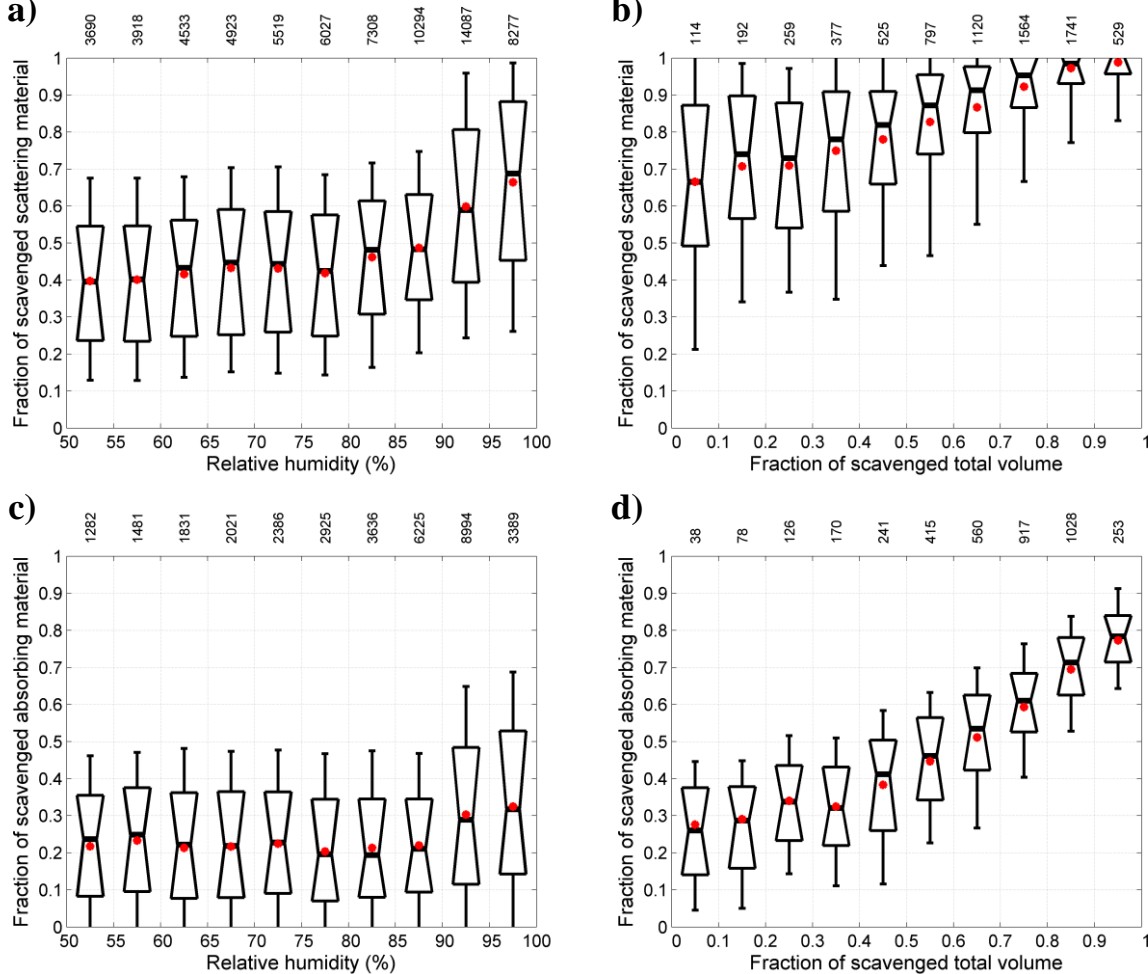

**Figure 4: Box plots of scavenging efficiency of scattering material as a function of (a) relative humidity and (b) fraction of scavenged total volume in cloud. Plots of scavenging efficiency of absorbing material as a function of (c) relative humidity and (d) fraction of scavenged total volume in cloud. Boxes similarly as in Fig. 3.**

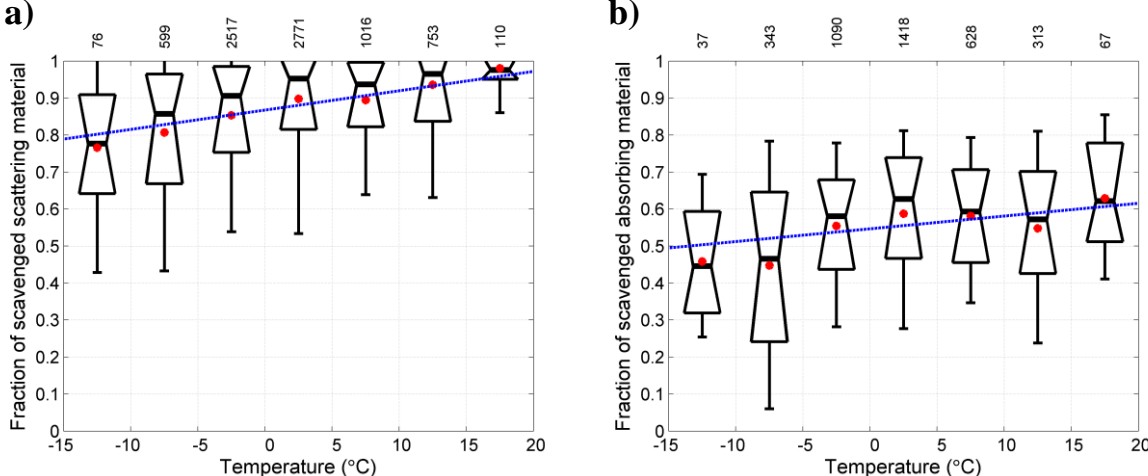

**Figure 5: Box plots of in cloud (a) scavenging efficiency of scattering material, and, (b) scavenging efficiency of absorbing material as a function of temperature. Boxes similarly as in Fig. 3.**

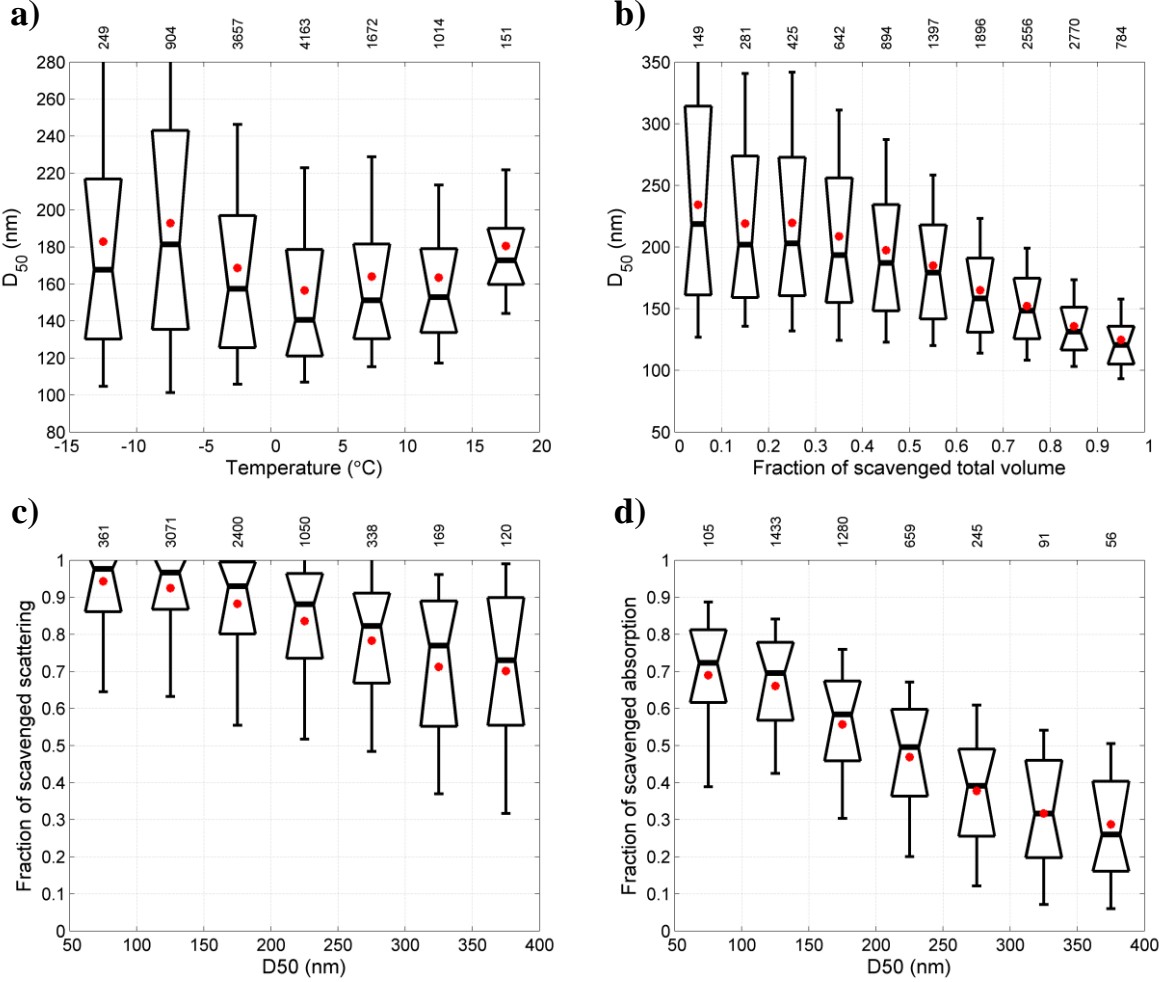

**Figure 6: $D_{50}$ as a function of (a) temperature and (b) fraction of scavenged total volume. Additionally fraction of scavenged scattering (c) and absorption (d) as a function of $D_{50}$. All plots during in cloud. Boxes similarly as in Fig. 3.**

