# Peer review of "Observations on aerosol optical properties and scavenging during cloud events"

_Atmospheric Chemistry and Physics, 2020_

## Referee Comment (RC1) · Anonymous Referee #1 · 29 Sep 2020

Review of "Observations on aerosol optical properties and scavenging during cloud events" by Ruuskanen et al.

This study reports on aerosol optical properties and scavenging during cloud events in Kuopio, Finland between October 2010 and November 2014. Measurements included aerosol size distributions and scattering coefficients, and absorption coefficient. The clouds examined were of a stratus type. The topic of wet scavenging is important and not well understood and therefore of relevance to this journal. The data collected were unique and of good value. There are scarce reports of the size dependent aerosol properties within the context of scavenging in stratus clouds. This work didn't use a CVI inlet but rather relied on the use and difference between separate inlets sampling interstitial and total aerosols.

[Figure]

Selected findings include: accumulation mode particle concentration being temperature dependent (higher with increasing T) presumed to be due to biogenic aerosol; in-cloud data show that scattering and absorbing materials scavenging efficiencies have a slight increasing T dependence; higher fraction of absorbing materials (relative to scattering materials) at smaller sizes in cloud. In terms of the presentation, I found the reporting of the results to be a bit confusing at times. English editing is required as many sentences had writing issues that made it hard to understand. I have some major comments below that most certainly should be addressed along with more specific comments at the end. The paper has potential to be published in ACP but needs major revisions.

Major Comments: Have the sampling inlets been characterized? Provide more information about their performance and provide also relevant references to detail their construction and how they perform in terms of size cuts and transmission efficiencies.

There is a lack of meteorological and thermodynamic data provided for the study site to help with data interpretation. I suggest adding data for boundary layer height for the sample times in addition to other relevant weather data such as rain amounts. Figure 1 is somewhat helpful but it doesn't show how much sampling occurred in the broad range of October 2010 and November 2014 and it doesn't show characteristics associate with boundary layer height or cloud types/properties.

Some discussion about the location is needed such as population characteristics and local/regional pollution sources.

When discussing trends in the data such as comparing results as a function of temperature and humidity, there is very little discussion of statistical tests to prove there is any significant difference or trend. Please provide more detailed statistical analysis for all trends/differences discussed in paper.

More discussion about the types of clouds studied is needed. What were there characteristics in terms of liquid water amount, depth, base heights, etc? More information is

needed, and if the authors did not measure such information, try to obtain information from reanalysis or remote sensing products. This is important to put into better context what types of clouds were examined. Simply saying "stratus" in my view is insufficient.

Specific Comments: I suggest adding a map figure showing the sample site in relation to its surroundings.

Page 1, Line 25: "the" should be "there"

Page 2, line 16: I suggest a reference after "microphysical processing" such as:

Ervens, B., et al. (2011), Secondary organic aerosol formation in cloud droplets and aqueous particles (aqSOA): A review of laboratory, field and model studies, Atmos. Chem. Phys., 11, 11,069–11,102, doi:10.5194/acp-11-11069-2011.

Wonaschuetz, A. et al. Aerosol and gas re-distribution by shallow cumulus clouds: An investigation using airborne measurements. J Geophys Res-Atmos 117, doi:10.1029/2012jd018089 (2012).

Page, Line 36-40: the use of these two inlets makes sense in the context of this study but are there any limitations to this method? Would a CVI be beneficial in such studies? I recommend a sentence or two to address this point for those planning to do similar studies in the future. Figure 1: what are the bars and what are the colored curves?

Page 5, line 33: provide some kind of evidence or support for this claim about lack of biogenic emissions.

Page 6, Line 9-12: this sentence doesn't make sense as written.

Page 7, Line 31: are these slopes even important or significant? There is a lack of any statistical analysis.

---

## Referee Comment (RC2) · Anonymous Referee #2 · 4 Oct 2020

GENERAL REMARKS

The manuscript reports results from a multi-annual study on aerosol optical properties and scavenging effects, observed at the Puijo measurement station in Kuopio, Finland. Observed properties include number concentrations of different aerosol modes, size distributions, scattering and absorption coefficients. The authors investigate the impact of environmental parameters (temperature, relative humidity, in-cloud and clear sky conditions) on the scavenging and wet deposition of various aerosol parameters. The experimental part of the study is very well designed and carefully conducted. The resulting data are of high quality and of high relevance for the investigation of the aerosol indirect effect because the data set covers a long period in time and thus a large variety of weather situations and atmospheric conditions. The interpretation

of the presented data, however, remains largely on the level of describing observed phenomena, whereas modelling studies for quantitative analyses are lacking.

In summary, the topic of the manuscript fits well into the scope of the journal. The manuscript can be accepted for publication in ACP after major revisions have been considered which are specified in the following.

SPECIFIC COMMENTS

1. The authors discuss the influence of environmental parameters like temperature and relative humidity on the properties of the sampled aerosol, and in particular of the scavenging efficiency. In the abstract and also later in the main text of the manuscript, the authors discuss the impact of air temperature on the observations just as if there is a direct dependency of temperature on the observed properties. However, particularly the impact of air temperature on the observed effects is only of indirect nature, since the aerosol population and constituents change with season and thus with temperature because of changing sources. An even better wording could be to describe the link between temperature and the observed effects as a correlation instead of a dependency. This fact should be clearly stated because in the current manuscript it reads like there is a clear temperature-dependence on aerosol properties like number concentrations etc.; see e.g. lines 19 to 22 of the abstract. The same is probably true for the impact of relative humidity since scavenging efficiencies at the same relative humidity level may change between, e.g., spring and fall conditions with different aerosol chemical compositions. Again, the effect of relative humidity on the hygroscopic behaviour of the aerosols is not only related to the level of relative humidity but also to the difference in chemical composition.

2. In its current version, the analyses presented in Figures 3 to 6 may suffer from pooling different aerosol chemical compositions and thus different hygroscopic growth behaviour into single bins for temperature and relative humidity. The authors describe this effect on page 6 lines 12 to 25 for the data shown in Figure 4, but not in a quantitative manner. The missing quantification however, makes the data less valuable for modelling studies since key properties are missing in the analysis. To overcome this limitation, it might be worthwhile to investigate, e.g., the contributions of various parameters on the variability of the fraction of scavenged absorbing material at -7°C (Figure 5b). In the current analysis, this fraction is centred at 0.45 with a P10 value below 0.1 and a P90 value close to 0.8. A similar exercise could be conducted for most of the other analyses.

3. In the introduction section (page 2, lines 13 to 25), the authors describe the interaction of aerosol particles with water vapour. This section requires rewriting for several reasons. The effect of hygroscopic growth at relative humidity < 100%, and even more important, the effect of cloud condensation nucleus activation is not related to microphysical processing but to water uptake by hygroscopic material. A more precise description is needed here. Later in this paragraph, the authors discuss that due to the different hygroscopic properties which favour scavenging of water-soluble light-scattering material, light absorbing aerosol is enriched in cloud-processed air parcels compared to its initial state, the cooling effect of liquid-water clouds is reduced compared to the warming effect of light absorbing material. A more detailed description and references are needed here.

4. In Figure 5, the authors present a regression analysis of temperature dependence of the fractions of lights scattering and light absorbing material; see Section 3.2.2. The results of this regression analysis are also listed in the main conclusions of the manuscript. The authors explained that the errors of observations were taken into account by performing a Deming regression analysis. The applied method is a suitable choice, but the statistical significance of the obtained results needs to be discussed.

MINOR ISSUES

Page 2, line 39: The authors state that the effect of clouds and precipitation on aerosol properties has been studied in a few campaigns. However, there have been many field

campaigns conducted on this topic, which is also reflected in the list of references given in the manuscript. An adequate restatement is requested.

Page 5, line 6: The minimum diameter for the total aerosol should be stated here.

Page 6, line 35: At some positions in the manuscript an article is missing, e.g., "First, sampling system with . . ." should read "First, a sampling system with . . .". Checking the manuscript text is recommended.

Page 7, line 12: a comma should be added after "Third".
* * *

---

## Author Comment (AC1) · 15 Nov 2020

**Referee #1**

This study reports on aerosol optical properties and scavenging during cloud events in Kuopio, Finland between October 2010 and November 2014. Measurements included aerosol size distributions and scattering coefficients, and absorption coefficient. The clouds examined were of a stratus type. The topic of wet scavenging is important and not well understood and therefore of relevance to this journal. The data collected were unique and of good value. There are scarce reports of the size dependent aerosol properties within the context of scavenging in stratus clouds. This work didn't use a CVI inlet but rather relied on the use and difference between separate inlets sampling interstitial and total aerosols.

Selected findings include: accumulation mode particle concentration being temperature dependent (higher with increasing T) presumed to be due to biogenic aerosol; in-cloud data show that scattering and absorbing materials scavenging efficiencies have a slight increasing T dependence; higher fraction of absorbing materials (relative to scattering materials) at smaller sizes in cloud. In terms of the presentation, I found the reporting of the results to be a bit confusing at times. English editing is required as many sentences had writing issues that made it hard to understand. I have some major comments below that most certainly should be addressed along with more specific comments at the end. The paper has potential to be published in ACP but needs major revisions.

> We thank Referee #1 for carefully reading the manuscript and providing valuable suggestions to improve the manuscript. Our point-by-point replies to the specific comments are given below. The referee comments are highlighted in blue and our responses are highlighted in black and indented. Excerpts from the revised manuscript are in Italics.

Major Comments:

Have the sampling inlets been characterized? Provide more information about their performance and provide also relevant references to detail their construction and how they perform in terms of size cuts and transmission efficiencies.

> The interstitial inlet is a commercial impactor manufactured by Digitel-AG (http://www.digitel-ag.com/de/en/products/low-volume-sampler/low-volume-inlets/) and has been characterized and calibrated by the manufacturer. The total air inlet at Puijo is similar to that used at the high-alpine site Jungfraujoch and has been characterized by Weingartner et al. (1999). We have characterized the inlets and sampling lines and their performance by comparing the particle size distributions in the interstitial and total sampling lines during a cloud-free weather and found that the aerosol size distributions in the interstitial and total air sampling lines, after particle loss corrections, match well in the measured range of 3–800 nm. We will add, accordingly, a more detailed description of the twin inlet and sampling line system in the revised manuscript.

There is a lack of meteorological and thermodynamic data provided for the study site to help with data interpretation. I suggest adding data for boundary layer height for the sample times in

addition to other relevant weather data such as rain amounts. Figure 1 is somewhat helpful but it doesn't show how much sampling occurred in the broad range of October 2010 and November 2014 and it doesn't show characteristics associate with boundary layer height or cloud types/properties.

For helping with the data interpretation, we will add into the supplement the following series of figures showing the monthly averages of temperature, relative humidity, visibility, and rainfall measured at Puijo during the sampling period (October 2010 – November 2014):

[Figure]

Unfortunately, we don't have reliable observations for cloud thickness and boundary layer height that could be used in this manuscript (see also comment about cloud types below).

The monthly coverages of valid data for the nephelometer, MAAP, and DMPS will be added to Figure 1 as Fig. 1b:

[Figure]

In order to give contrast for the valid data coverages, we will also add the average raw data coverage indicating the working performance of the instruments during the whole sampling period (October 2010 – November 2014) which are 64% for the nephelometer, 95% for the MAAP, and 95% for the DMPS.

Some discussion about the location is needed such as population characteristics and local/regional pollution sources.

A map (please see also reply to the specific comment #1) and short description about the location surroundings will be added into the revised manuscript:

*About 91 000 people live in the Kuopio urban area (see map in S1). Puijo hill (elevation 150 m) and the measurement station is located ca 2 km NW from the city centre. There are a few well-known local sources: the district heating plant 3.5 km SE, the paper mill 5 km NE, and a highway across the city. The distances and directions are in relation to the measurement station. Several residential areas are located nearby, but mostly east and south from the station.*

When discussing trends in the data such as comparing results as a function of temperature and humidity, there is very little discussion of statistical tests to prove there is any significant difference or trend. Please provide more detailed statistical analysis for all trends/differences discussed in paper.

The 95% confidence levels were calculated, and they show statistical significance. The section discussing the statistical tests will be reformulatd as:

*The slopes for scavenging efficiencies were calculated to be approximately 0.0052 (95% CL for slope: 0.0042-0.0062) for scattering (Fig. 5a) and 0.0034 (0.0022-0.0046) for absorption (Fig. 5b). Which means that the effects are statistically significant and corresponding to approximately 5 percentage unit change for scattering and 3 for the absorbing material in the scavenged fraction at 10°C change in temperature, respectively.*

More discussion about the types of clouds studied is needed. What were there characteristics in terms of liquid water amount, depth, base heights, etc? More information is needed, and if the authors did not measure such information, try to obtain information from reanalysis or remote sensing products. This is important to put into better context what types of clouds were examined. Simply saying "stratus" in my view is insufficient.

The station is located in 224 m above the local lake level and therefore only low-level clouds are observed. The criteria for the presence of a cloud is that the average visibility is less than 200 m, which excludes all broken clouds, and only relatively persistent stratified clouds are included in the observations, with possible occasional fogs. Due to the visibility criterion, also the cases with lowest liquid water contents (LWC) are filtered out. We do not have independent LWC observations covering the whole period, but when available, it has varied between 0.01 g×m$^{-3}$ and 0.27 g×m$^{-3}$ (Portin, H., Leskinen, A., Hao, L., Kortelainen, A., Miettinen, P., Jaatinen, A., Laaksonen, A., Lehtinen, K. E. J., Romakkaniemi, S. and Komppula, M.: The effect of local sources on particle size and chemical composition and their role in aerosol–cloud interactions at Puijo measurement station, Atmospheric Chemistry and Physics, 14, 6021-6034, 2014).

Unfortunately, we do not have reliable observations to characterize the cloud thickness from continuous observations, but we will carry out re-analysis or our data set in more detail, e.g., by inspecting the sensitivity of screening for precipitating cases, for which we have used a lower limit of 0.2 mm/h for the precipitation intensity. The screening for precipitating cases limits the thickness of clouds to some degree and removes the cases where lower cloud is seeded by precipitation from above. Furthermore, we did not have active LWC measurements to confirm the cloud types, but cloud base height information, obtained from a ceilometer 2 km south of the Puijo station, could be used in interpreting the data in more detail. The same uncertainty is related to possible presence of clouds above the observed clouds, which could affect the radiative cooling which is likely the main driver of boundary layer dynamics during the fall season when most of the clouds are observed. This is a topic which we will study in more detail based on the data from a currently ongoing measurement campaign, where we also use a cloud radar for better quantification of cloud thicknesses and different cloud layers. In the same context we can also test the applicability of the re-analysis for cloud characterization at Puijo station. The effect of unknown cloud thickness will be discussed in more detail in the revised manuscript. We thank the reviewer for this excellent idea.

Specific Comments: I suggest adding a map figure showing the sample site in relation to its surroundings

The following map will be added into the supplement:

[Figure]

Page 1, Line 25: "the" should be "there"

Corrected as suggested:

*This analysis indicated that there is a higher fraction of absorbing material, typically black carbon, in smaller sizes so that at least 20–30% of interstitial particles within clouds consist of absorbing material*

Page 2, line 16: I suggest a reference after "microphysical processing" such as:

Ervens, B., et al. (2011), Secondary organic aerosol formation in cloud droplets and aqueous particles (aqSOA): A review of laboratory, field and model studies, Atmos. Chem. Phys., 11, 11,069–11,102, doi:10.5194/acp-11-11069-2011.

Wonaschuetz, A. et al. Aerosol and gas re-distribution by shallow cumulus clouds: An investigation using airborne measurements. J Geophys Res-Atmos 117,doi:10.1029/2012jd018089 (2012).

We have inserted the suggested references into the revised manuscript.

Page, Line 36-40: the use of these two inlets makes sense in the context of this study but are there any limitations to this method? Would a CVI be beneficial in such studies? I recommend a sentence or two to address this point for those planning to do similar studies in the future.

It is true that a CVI would be beneficial for these kinds of studies because it removes the interstitial aerosol particles from the total aerosol, enabling direct studies of the activated fraction. The two inlet system, in turn, relies on measuring the particle properties in the total and interstitial aerosols separately and by subtracting the interstitial from the total, the properties of the activated aerosol can be studied, which may lead to misinterpretations, particularly if the studied aerosol property is not of an additive nature.

Furthermore, small discrepancies may arise from the delay between the total and interstitial aerosol measurement, particularly in fast changing situations. All in all, both systems have their pros and cons, and the choice should be made bearing in mind the needs or restrictions of the site and sampling setup. Of course, and if possible, both systems can be used in parallel, which could then be used for intercomparison of their performances. We will add discussion about these points in the revised manuscript.

Figure 1: what are the bars and what are the colored curves?

Explanation for these will be added to the caption:

*The average temperature is presented with black curve and the daily minimum and maximum temperatures are presented with the curves associated with blue and red areas, respectively. The occurrence of cloud events is presented with grey bars.*

Page 5, line 33: provide some kind of evidence or support for this claim about lack of biogenic emissions.

Emissions from biogenic sources have been measured, e.g., in Hyytiälä, Finland by Heikkinen et al. (2020), which reference will be inserted into the text.

*Heikkinen, L., Äijälä, M., Riva, M., Luoma, K., Dällenbach, K., Aalto, J., Aalto, P., Aliaga, D., Aurela, M., Keskinen, H., Makkonen, U., Rantala, P., Kulmala, M., Petäjä, T., Worsnop, D., and Ehn, M.: Long-term sub-micrometer aerosol chemical composition in the boreal forest: inter- and intra-annual variability, Atmos. Chem. Phys., 20, 3151–3180, https://doi.org/10.5194/acp-20-3151-2020, 2020*

Page 6, Line 9-12: this sentence doesn't make sense as written.

Thank you for the comment, the sentence has been rewritten for clarity:

*The compounds which are dominantly in the largest particles are scavenged at high relative humidities even without any clouds present. These large particles are not considered to be cloud scavenged. Meanwhile, the scavenging of compounds in the smaller particle sizes is affected less by the increase in the relative humidity.*

Page 7, Line 31: are these slopes even important or significant? There is a lack of any statistical analysis.

The significance levels were calculated and will be added to the text (see also the reply to the corresponding major comment about confidence levels):

*The slopes for scavenging efficiencies were calculated to be approximately 0.0052 (95% CL for slope: 0.0042-0.0062) for scattering (Fig. 5a) and 0.0034 (0.0022-0.0046) for absorption (Fig. 5b). Which means that the effects are statistically significant and corresponding to approximately 5 percentage unit change for scattering and 3 for the absorbing material in the scavenged fraction at 10°C change in temperature, respectively.*

---

## Author Comment (AC2) · 15 Nov 2020

**Referee #2**

GENERAL REMARKS

The manuscript reports results from a multi-annual study on aerosol optical properties and scavenging effects, observed at the Puijo measurement station in Kuopio, Finland. Observed properties include number concentrations of different aerosol modes, size distributions, scattering and absorption coefficients. The authors investigate the impact of environmental parameters (temperature, relative humidity, in-cloud and clear sky conditions) on the scavenging and wet deposition of various aerosol parameters. The experimental part of the study is very well designed and carefully conducted. The resulting data are of high quality and of high relevance for the investigation of the aerosol indirect effect because the data set covers a long period in time and thus a large variety of weather situations and atmospheric conditions. The interpretation of the presented data, however, remains largely on the level of describing observed phenomena, whereas modelling studies for quantitative analyses are lacking. In summary, the topic of the manuscript fits well into the scope of the journal. The manuscript can be accepted for publication in ACP after major revisions have been considered which are specified in the following.

> We thank Referee #2 for carefully reading the manuscript and providing valuable suggestions to improve the manuscript. Our point-by-point replies to the specific comments are given below. The referee comments are highlighted in blue with our responses in black and indented. The excerpts from the revised manuscript are in Italics and the font size is smaller.

SPECIFIC COMMENTS

1. The authors discuss the influence of environmental parameters like temperature and relative humidity on the properties of the sampled aerosol, and in particular of the scavenging efficiency. In the abstract and also later in the main text of the manuscript, the authors discuss the impact of air temperature on the observations just as if there is a direct dependency of temperature on the observed properties. However, particularly the impact of air temperature on the observed effects is only of indirect nature, since the aerosol population and constituents change with season and thus with temperature because of changing sources. An even better wording could be to describe the link between temperature and the observed effects as a correlation instead of a dependency. This fact should be clearly stated because in the current manuscript it reads like there is a clear temperature-dependence on aerosol properties like number concentrations etc.; see e.g. lines 19 to 22 of the abstract. The same is probably true for the impact of relative humidity since scavenging efficiencies at the same relative humidity level may change between, e.g., spring and fall conditions with different aerosol chemical compositions. Again, the effect of relative humidity on the hygroscopic behaviour of the aerosols is not only related to the level of relative humidity but also to the difference in chemical composition.

> We thank the referee for this very good comment and will revise the manuscript accordingly in order to emphasize the indirect nature of the environmental parameters, such as temperature, on the studied aerosol properties. We chose temperature as the main environmental parameter because it is, despite of its indirect nature, the easiest parameter to be used in the analysis and in comparison with model calculations. Furthermore, similar representation has been used in several previous studies, which enabled us to carry out straightforward comparison with the literature.

2. In its current version, the analyses presented in Figures 3 to 6 may suffer from pooling different aerosol chemical compositions and thus different hygroscopic growth behaviour into single bins for temperature and relative humidity. The authors describe this effect on page 6 lines 12 to 25 for the data shown in Figure 4, but not in a quantitative manner. The missing quantification however, makes the data

less valuable for modelling studies since key properties are missing in the analysis. To overcome this limitation, it might be worthwhile to investigate, e.g., the contributions of various parameters on the variability of the fraction of scavenged absorbing material at -7◦C (Figure5b). In the current analysis, this fraction is centred at 0.45 with a P10 value below 0.1and a P90 value close to 0.8. A similar exercise could be conducted for most of the other analyses.

> We agree with the referee that pooling too much data causes variation in the analysis results and that more detailed information would increase the value of the presented data. Unfortunately, we do not have, e.g., long-term mass spectrometry or updraft measurements at the cloud base which would help us in investigating the effect of aerosol chemistry and mixing state on the observed D50 values in more detail. However, we consider that providing average BC scavenging efficiencies and connecting it with cloud dynamics through the calculated D50 values provides a valuable data set for model evaluation purposes when low altitude clouds are considered.

3. In the introduction section (page 2, lines 13 to 25), the authors describe the interaction of aerosol particles with water vapour. This section requires rewriting for several reasons. The effect of hygroscopic growth at relative humidity < 100%, and even more important, the effect of cloud condensation nucleus activation is not related to microphysical processing but to water uptake by hygroscopic material. A more precise description is needed here. Later in this paragraph, the authors discuss that due to the different hygroscopic properties which favour scavenging of water-soluble light-scattering material, light absorbing aerosol is enriched in cloud-processed air parcels compared to its initial state, the cooling effect of liquid-water clouds is reduced com-pared to the warming effect of light absorbing material. A more detailed description and references are needed here.

> We will rephrase the paragraph in order to state more clearly that black carbon (BC) resides typically in relatively small particles, which activate unlikely to cloud droplets and thus are also less likely to be wet scavenged, whereas the more hygroscopic fraction is typically found in larger particles and is more likely scavenged through cloud processes. This holds at least in low altitude clouds where below cloud washout is relatively less important than in cloud scavenging. Thus, the lifetime of BC is expected to be longer.

4. In Figure 5, the authors present a regression analysis of temperature dependence of the fractions of lights scattering and light absorbing material; see Section 3.2.2.The results of this regression analysis are also listed in the main conclusions of the manuscript. The authors explained that the errors of observations were taken into account by performing a Deming regression analysis. The applied method is a suitable choice, but the statistical significance of the obtained results needs to be discussed.

> We introduced the confidence levels and added a short statement about the significance:
>
> *The slopes for scavenging efficiencies were calculated to be approximately 0.0052 (95% CL for slope: 0.0042-0.0062) for scattering (Fig. 5a) and 0.0034 (0.0022-0.0046) for absorption (Fig. 5b). Which means that the effects are statistically significant and corresponding to approximately 5 percentage unit change for scattering and 3 for the absorbing material in the scavenged fraction at 10°C change in temperature, respectively.*

MINOR ISSUES

Page 2, line 39: The authors state that the effect of clouds and precipitation on aerosol properties has been studied in a few campaigns. However, there have been many field campaigns conducted on this topic, which is also reflected in the list of references given in the manuscript. An adequate restatement is requested.

We reformulated the sentence which currently reads as:

*The effect of clouds or precipitation on aerosol optical properties has also been studied in several campaigns (Zhang et al., 2012; Berkowitz et al., 2011; Hyvärinen et al., 2011; Chaubey et al., 2010; Marcq et al., 2010; Yamagata et al., 2009; Cozic et al, 2007; Latha et al., 2005) and by conducting model calculations (e.g., Browse et al., 2012; Croft et al., 2009).*

Page 5, line 6: The minimum diameter for the total aerosol should be stated here.

We added the minimum diameter which is the same for both measurement lines.

*The total number concentration ($N_{tot}$) is defined as the number concentration of all particles ranged from 3 or 7 nm to 800 nm (see section 2.2).*

Page 6, line 35: At some positions in the manuscript an article is missing, e.g., "First, sampling system with..." should read "First, a sampling system with...". Checking the manuscript text is recommended.

We will check the grammar throughout the revised manuscript as suggested.

Page 7, line 12: a comma should be added after "Third".

The sentence has been rephrased slightly for clarification:

*The third and last possible source of dispersion resides in long term measurements.*